# NonToxic Silver/Poly-1-Vinyl-1,2,4-Triazole Nanocomposite Materials with Antibacterial Activity

**DOI:** 10.3390/nano10081477

**Published:** 2020-07-28

**Authors:** Irina A. Shurygina, Galina F. Prozorova, Irina S. Trukhan, Svetlana A. Korzhova, Tatiana V. Fadeeva, Alexander S. Pozdnyakov, Nataliya N. Dremina, Artem I. Emel’yanov, Nadezhda P. Kuznetsova, Michael G. Shurygin

**Affiliations:** 1Irkutsk Scientific Center of Surgery and Traumatology, 1 Bortsov Revolutsii St., 664003 Irkutsk, Russia; PREDEL4@yandex.ru (I.S.T.); fadeeva05@yandex.ru (T.V.F.); drema76@mail.ru (N.N.D.); MShurygin@gmail.com (M.G.S.); 2A.E. Favorsky Irkutsk Institute of Chemistry, Siberian Branch, Russian Academy of Sciences, 1 Favorsky Street, 664033 Irkutsk, Russia; prozorova@irioch.irk.ru (G.F.P.); korzhova@irioch.irk.ru (S.A.K.); pozdnyakov@irioch.irk.ru (A.S.P.); emelyanov@irioch.irk.ru (A.I.E.); nkuznetsova@irioch.irk.ru (N.P.K.)

**Keywords:** silver nanoparticles, poly-1-vinyl-1,2,4-triazole, nontoxic nanocomposite, antimicrobial activity

## Abstract

Novel silver/poly-1-vinyl-1,2,4-triazole nanocomposite materials—possessing antimicrobial activity against Gram-positive and Gram-negative bacteria—have been synthesized and characterized in the solid state and aqueous solution by complex of modern physical-chemical and biologic methods. TEM-monitoring has revealed the main stages of microbial cell (*E. coli*) destruction by novel nanocomposite. The concept of direct polarized destruction of microbes by nanosilver proposed by the authors allows the relationship between physicochemical and antimicrobial properties of novel nanocomposites. At the same time, it was shown that the nanocomposite was nontoxic to the fibroblast cell culture. Thus, the synthesized nanocomposite combining antibacterial activity against Gram-positive and Gram-negative bacteria as well as the absence of toxic effects on mammalian cells is a promising material for the development of catheters, coatings for medical devices.

## 1. Introduction

Unflagging interest of researchers in silver nanoparticles (NPs) and their composites derives from ever-increasing application of these multipurpose nanomaterials in avalanche-like developing modern biomedical and other interdisciplinary nanotechnologies [1,2,3]. In this regard, not only fundamental effects of silver NPs, but also diverse specific physical–chemical and biologic properties of specific polymeric matrices stabilizing silver NPs are of importance. These matrices are used for the production of functional silver nanocomposites (where they are incorporated to) possessing a set of remarkable properties which are mainly due to these polymers. Therefore, the design of novel Ag(0)-nanocomposites based on original polymers as well as the investigation of properties of these polyfunctional hybrid metal–polymer nanomaterials remains an urgent challenge.

It is well-known that most metal-containing NPs are toxic to humans which raise concerns about these engineered particles [4].

Unfortunately, many silver nanocomposites having unique antibacterial properties are also toxic to mammalian cells [5,6,7,8,9,10]. One of the approaches allowing to reduce the silver nanoparticles (AgNPs) cytotoxicity is green synthesis application [11,12,13]. At the same time, the use of various matrices for stabilization of nanoparticles also promotes to decrease the cytotoxicity of silver nanoparticles [14,15].

Water-soluble poly-1-vinyl-1,2,4-triazole (PVT) is a nontoxic (LD_50_ > 5000 mg/kg) synthetic heterocyclic polymer with chemical and thermal stability and wide functionality [16]. The combination of biocompatibility, controlled solubility and other valuable properties of PVT and its copolymers makes them particularly promising materials for biomedicine; (co)polymers of 1-vinyl-1,2,4-triazole exhibit a high stabilizing ability during forming of the polymer silver-containing nanocomposites that have strong antimicrobial activity [17].

However, despite the tremendous flow of fundamental information on biologic activity of silver ions and nanoparticles, in particular, on their antimicrobial action (summarized in recent reviews and manuscripts) the complex interaction of microbial culture and silver nanocomposite in whole as well as with taking into account the nature of specific functional nano-stabilizing polymeric matrix remains the key problem.

In this study, to analyze nanoparticles cytotoxicity, we used a cell culture model because it is known that in general the cell line is more than 1000 times greater sensitive than the in vivo models [18]. At the same time, normal fibroblast culture can be considered more informative tool to detect the mechanisms of metallic NPs action since AgNPs were found to be the most cyto–genotoxic NPs to fibroblasts [4].

In this study, we report on the synthesis of new water-soluble nanocomposites with AgNPs, stabilized by biocompatible polymer PVT and study their physicochemical, structural, optical, thermal properties and biologic activity. We discuss the results of studying the antimicrobial activity of the obtained nanocomposites against both Gram-positive and Gram-negative bacteria, the main stages of the destruction of microbial cells (*E. coli*) by a new nanocomposite monitored by TEM and the degree of toxic effect of the nanocomposite on the fibroblast cell culture.

## 2. Materials and Methods

### 2.1. Materials

The starting monomer 1-vinyl-1,2,4-triazole (VT) was synthesized according to the published method [19] using 1,2,4-triazole from Sigma-Aldrich (St. Louis, MO, USA). PVT was synthesized by radical polymerization of VT and was used as the stabilizing agent. Azobisisobutyronitrile (AIBN) was acquired from Merck KGaA (Darmstadt, Germany) and was used as polymerization initiator. Silver nitrate (AgNO_3_) was purchased from Sigma-Aldrich (St. Louis, MO, USA) and used as metallic precursor of silver nanoparticles (AgNPs). Sodium borohydride (NaBH_4_) and formaldehyde (CH_2_O) were obtained from Sigma-Aldrich (St. Louis, MO, USA) and used as strong and intermediate reducing agents, respectively. AIBN, AgNO_3_, NaBH_4_ and CH_2_O reagents are used without further purification. Dimethylformamide (DMF) (PETRONIT, Yekaterinburg, Russia) was purified by distillation over calcium hydride. Deionized water (H_2_O) was used to prepare solutions.

Dulbecco’s modified Eagle’s medium (DMEM), fetal bovine serum (FBS), antibiotic/antimycotic (10,000 U/mL of penicillin, 10 µg/mL of streptomycin and 25 µg/mL of amphotericin B), collagenase, glutaraldehyde, fluorescein diacetate (FDA) (Cat. F7378, Lot MKBR3002 V) and propidium iodide (PI) (Cat. P4864, Lot MKCC9922) were obtained from Sigma-Aldrich Co. (St. Louis, MO, USA). Mueller–Hinton agar was purchased from HiMedia Laboratories Pvt. Limited (Mumbai, India). Bacterial strains used for the experiments were obtained from ATCC (Manassas, VA, USA). Two Gram-positive reference strains for antibiotic testing, namely *Staphylococcus aureus* (strain code: ATCC 25923), *Enterococcus faecalis* (ATCC 29212) and three Gram-negative strains, namely *Klebsiella pneumoniae* (ATCC 700603), *Escherichia coli* (ATCC 25922) and *Pseudomonas aeruginosa* (ATCC 27,853) were used for the antibacterial evaluation.

### 2.2. Synthesis of PVT

PVT was synthesized by free radical polymerization of VT in DMF and argon atmosphere in the presence of AIBN initiator (1% to the monomer mass) at 60 °C for 6 h. (Scheme 1).

The resulting polymer (yield was equal to 92%, average molecular weight of 60 kDa) was fractionated by selective dissolution in DMF and precipitation in acetone to obtain seven fractions with a molecular weight of 15–70 kDa. A polymer with a molecular weight of 26 kDa was well dissolved in water and was subsequently used as a stabilizing matrix in the synthesis of silver-containing nanocomposites.

### 2.3. Synthesis of AgNPs Nanocomposites

The AgNPs incorporated in the stabilizing polymer matrix were formed by chemical reduction of silver ions from silver nitrate using sodium borohydride or formaldehyde as a stronger and less strong reducing agent, respectively. The reactions were carried out in an aqueous solution of PVT at room temperature for 10 h. As a result, new hybrid organo-inorganic polymer silver-containing nanocomposites **1**–**6** were obtained. The content of AgNPs in the polymer matrix was varied by changing the ratio of the amount of the initial metal precursor to the polymer.

AgNPs nanocomposite **1**. An aqueous solution (5 mL) of AgNO_3_ (1.0 mmoL) was slowly dropwise added to an aqueous solution (40 mL) of PVT (50.0 mmoL). The reaction mixture was stirred for 30 min and then NaBH_4_ (1.0 mmoL) was gradually added to it. The reaction was carried out at room temperature for 10 h with vigorous stirring. After completion of the reaction, the reaction mixture, which turned dark brown, was precipitated into ethanol. The resulting product was filtered and dialyzed through a 5-kDa membrane (MFPI, Cellu-Sep H1) for 72 h. The obtained nanocomposite was isolated by freeze-drying. Thus, the nanocomposite was synthesized in the form of a dark brown powder with a silver content of 5%.

AgNPs nanocomposites **2**–**5** were synthesized according to the above method, but with a different ratio of AgNO_3_, PVT and NaBH_4_ (Table 1).

AgNPs nanocomposite **6** was synthesized similarly to the synthesis method of nanocomposite **1**, but formaldehyde was used as a reducing agent.

### 2.4. Instruments for Characterization of Polymer and AgNPs Nanocomposites

Elemental analysis was carried out on a Thermo Scientific Flash 2000 Analyzer (Thermo Fisher Scientific, Waltham, MA, USA). The molecular weight of PVT was determined by gel permeation chromatography at 50 °C using Shimadzu LC-20 prominence system (Shimadzu Corporation, Kyoto, Japan) fitted with a differential refractive index detector Shimadzu RID-20A (Shimadzu Corporation, Kyoto, Japan) and column Agilent PolyPore 7.5 × 300 mm (PL1113-6500) (Santa Clara, CA, USA). DMF solution was used as the eluent at the flow rate of 1 mL/min. Dissolution of samples was performed at 50 °C for 24 h with stirring. A number of polystyrene standards were used for calibration. Fourier transform infrared (FTIR) spectroscopy was used to study the molecular structure of PVT and AgNPs nanocomposites. FTIR spectra of solid compounds were recorded in the range of 550–4000 cm^−1^ on a Vertex 70 spectrometer (Bruker Corporation, Billerica, MA, USA). A UV-Vis spectrophotometer Lambda 35 (PerkinElmer, Inc., Waltham, MA, USA) was used to study the process of AgNPs formation in a PVT polymer matrix. UV-visible spectra were recorded in the wavelength range of 200–750 nm. Using the method of atomic absorption analysis on an AAnalyst 200 instrument (PerkinElmer, Inc., Waltham, MA, USA), the silver content in the synthesized polymer silver-containing nanocomposites was determined. A Leo 906E transmission electron microscope (Carl Zeiss AG, Oberkochen, Germany) was used to study the morphology of nanocomposites, the size of the resulting AgNPs and the nature of their distribution in the polymer matrix. The synthesized nanocomposites were characterized by X-ray diffraction, which carried out by X-ray powder diffractometer D8 Advance (Bruker Corporation, Billerica, MA, USA) using Cu-irradiation. The zeta potential and the hydrodynamic radius of the nanoparticles were determined using a Zetasizer Nano-ZS photon-particle analyzer (Malvern Instruments Ltd, Malvern, UK). The measurements were carried out in thermostated cuvettes at a temperature of 25 °C. We used 0.1-M NaNO_3_ solutions with 0.1 mg/mL of the test sample. The polymer and nanocomposites stability to thermal oxidative degradation was studied by thermogravimetric analysis and differential scanning calorimetry using a STA 449 Jupiter derivatograph (Netzsch, Selb, Germany). The measurements were carried out in the temperature range 25–700 °C at a heating rate of 10 °C/min in the air atmosphere. The weight of the samples was 5 mg.

### 2.5. Bacterial Culture

For experimental investigation of antimicrobial action of the obtained nanocomposites we used such microorganism as *E. coli* (ATCC 25922), *P. aeruginosa* (ATCC 27853), *K. pneumoniae* (ATCC 700603), *S. aureus* (ATCC 25923) and *E. faecalis* (ATCC 29212).

Suspensions of test organisms in a concentration of 0.5 McFarland standard (1.5 × 10^8^ CFU (colony forming unit)/mL) prepared from an agar culture in a sterile isotonic sodium chloride solution using a densitometer Densimat (Bio Merieux, Marcy-l’Étoile, France). The density of *E. coli* daily culture was decreased to 10^7^ CFU/mL and 10^5^ CFU/mL by the method of 10-fold serial dissolution for electron microscopic study.

### 2.6. Determination of Antibacterial Action of Nanocomposites

The minimum inhibitory concentration and minimum bactericidal concentration (MIC and MBC, respectively) of the synthesized nanocomposite containing 5.0% AgNPs was studied using a serial dilution method [20].

The stock solutions of the nanocomposites contained 1000 µg/mL. Under aseptic conditions, these solutions were used to obtain serial two-fold dilutions of substances in liquid nutrient medium in a number of eight to eleven tubes (1 mL volume) with a final microorganism concentration of 5 × 10^5^ CFU/mL. Antimicrobial activity was studied with dilution of samples in a range of 0.125–500 µg/mL. To obtain the necessary inoculum (5 × 10^5^ CFU/mL), 50 μL of bacterial suspension containing 10^6^ CFU/mL was added in every tube. The control tube contained 50 μL of the culture one of the tested strains into1 mL of broth without adding the substance. The experiments were carried out twice for each culture.

The inocula were incubated under common atmosphere at 35 °C for 18–24 h. The results were evaluated visually determining presence or absence of growth in the medium containing the studied compound in different concentrations. The last tube with growth retardation (clear broth) corresponded to the minimum inhibitory concentration of the substances with respect to a strain.

To determine cell viability, the cultures from all transparent tubes were inoculated on a solid medium (Mueller–Hinton agar). After incubation of inoculum at 35 °C, the lowest concentration of the test substance with no bacterial growth was determined. This concentration corresponds to as MBC.

### 2.7. Incubation of Nanocomposite with E. coli for Electron Microscopic Study

Three series of tests were performed to study the interaction of the nanocomposite with *E. coli* (ATCC 25922). The stock 0.05% solution of the silver nanocomposite was prepared in the phosphate buffer (pH 7.6).

In each series of tests, the microbial suspension of *E. coli* (0.1 mL) was introduced into a vial with nanocomposite solution to reach the total volume of 1.0 mL. The samples containing the phosphate–salt buffer instead of the nanocomposite solution were used as control. The inoculations were incubated at 37 °C for 1, 2 and 24 h (for first, second and third series of tests, respectively). After the corresponding incubation was completed, the quantitative plating on Mueller–Hinton agar was performed.

The microorganism samples were fixed by a 6.5% solution of glutaraldehyde in the phosphate–salt buffer (pH = 7.6) for 2 h. Then the samples were absorbed on nickel grids covered with evaporated carbon film and studied using TEM-410 (Philips Electron Optics, Eindhoven, The Netherlands) [5].

### 2.8. Primary Fibroblast Culture Isolation

Isolation of primary fibroblast culture from the omentum of mature Wistar rats (200 g) was carried out. Utilized animals were housed in accordance with the good laboratory practice (GLP) rules. The experiment was performed in accordance with the norms for the humane treatment of animals regulated by the International guidelines of the association for the assessment and accreditation of laboratory animal care and in concordance with the protocol approved by the Institutional Animal Care and Use Committee of the Irkutsk Scientific Center of Surgery and Traumatology (protocol N 2, 29.02.2016). All the operative interventions were carried out under aseptic conditions. Animals were anesthetized with an intramuscular injection of 2% Rometar (Bioveta a. s., Ivanovice na Hané, Czech Republic) at the dose of 0.2 mL/kg body weight.

The primary culture was obtained by fragmenting the excised omentum and disaggregating the tissue fragments at 37 °C in the solution containing 200 U/mL of collagenase, 2% antibiotic/antimycotic in DMEM. Collagenase activity in the suspension was inhibited by equal amount of DMEM containing 15% FBS and 1% antibiotic/antimycotic. Then the cells were twice washed in DMEM supplemented with 10% FBS, 1% antibiotic/antimycotic by centrifuging the suspension at 500 G for 5 min. Isolated fibroblasts were cultured in DMEM containing 10% FBS, 1% antibiotic/antimycotic at 37 °C, 80% humidity and 5% CO_2_ in the BioStation CT (Nikon Corporation, Tokyo, Japan ). To obtain a culture, fibroblasts were subcultured every 7 days [21].

### 2.9. Testing the Influence Studied Nanocomposite to the Fibroblast Culture

After the third passage the culture was treated with silver/PVT nanocomposite or PVT at the concentrations of 0.125 to 16 µg/mL. The cells that were not exposed to the active substances served as controls (an appropriate amount of DMEM was added to the culture). Research performed using BioStation CT.

Fibroblasts were cultured with active substances for 24 h, the incubated cells were photographed using the phase contrast method once every 12 h. Living cell were stained by fluorescein diacetate (FDA) in concentration of 2 μg/mL and dead cells were stained by propidium iodide (PI) in concentration of 10 μg/mL. BioStation CT was applied to visualize fluorescence staining. Intact fibroblasts were used as a control. Counting the number of living and dead cells was performed using the software product NIS-Elements AR, v. 5.00 (Nikon Instruments Inc., Melville, NY, USA). Statistical analysis was carried out in the programming environment R. For the analysis of variance, we used the Kruskal–Wallis test, a posteriori analysis was performed using the Tukey’s test.

## 3. Results

### 3.1. Characterization of PVT

PVT was used as a stabilizing matrix in the synthesis of silver-containing nanocomposites. The selected polymer with a molecular weight of 26 kDa is characterized by good solubility in water, DMF, DMSO, biocompatibility and nontoxicity (LD_50_ > 5000 mg/kg) [16].

In the solid state, the polymer is X-ray amorphous and is capable to form films with “cell” structure (Figure 1a). According to the data of thermogravimetric analysis, the polymer possesses high thermal stability, the decomposition temperature being above 330 °C. A study of photon correlation spectroscopy showed that in an aqueous solution each PVT molecule exists as a separate statistical globule with an average hydrodynamic diameter of 292 nm (polydispersity of 0.34) (Figure 1b).

### 3.2. Synthesis and Characterization of Polymeric AgNPs Nanocomposites

The mixing of silver nitrate and PVT aqueous solutions leads to strong gelation of the reaction mixture and the formation of highly viscous creamy gels. This is due to specific coordination crosslinking between linear PVT macromolecules, where silver ions act as coordinating crosslinking agents. In such reactions, silver ions have a coordination number of 2 (or higher) and can simultaneously form several coordination bonds with triazole rings, both intramolecular and intermolecular (Figure 2).

During subsequent intensive mixing and the addition of a reducing agent (sodium borohydride or formaldehyde), silver cations are reduced to a state of zero valence. The resulting metallic silver nanoparticles are stabilized in the PVT polymer matrix through coordination bonds between triazole rings and silver atoms on the surface of the nanoparticles. The reaction ends with the formation of brown sols, from which samples of nanocomposites **1**–**6** in the form of dark brown powders were isolated (Table 1).

According to elemental analysis and atomic absorption spectroscopy, the silver content in the obtained samples depends on the initial polymer / silver nitrate ratio and is 1.8–20.8% (Table 1). Nanocomposites are soluble in water, DMSO and DMF. The solubility of the obtained nanocomposites depends on their silver content. Samples with a metal content of up to 10% are well soluble in these solvents. With an increase in silver content above 10%, there is a loss of solubility of nanocomposites in all tested solvents. This can be explained by the fact that during the synthesis of nanocomposites, the formation of macromolecular globules occurs, induced by silver nanoparticles (Figure 3a). With an increase in the amount of silver, nanoparticles can participate not only in intramolecular coordination, but also in intermolecular crosslinking, which leads to the formation of insoluble crosslinked nanocomposites (Figure 3b). It results in the densification (“compression”) of macromolecular globules, a decrease in the hydrodynamic diameters of nanocomposite particles to 195 nm, compared with the diameter of macromolecules of the initial PVT (292 nm) (Figure 3b) and leads to loss of solubility of nanocomposites.

In optical absorption spectra of the nanocomposite aqueous solutions (unlike those of aqueous solutions of the starting silver nitrate and PVT), a characteristic band of plasmon resonance absorption of conductivity electrons of silver metal nanoparticles with a maximum at 412–426 nm (Figure 4a) appears that provides for intensive brown color of the nanocomposites.

In the X-ray diffractograms of the nanocomposites (Figure 4b), amorphous halo of the polymeric matrix and intensive reflexes of zero-valence metal silver are clearly differentiated. The latter were identified by the comparison of the values of interplanar spacing and relative intensities with standard values for metal silver. An X-ray diffraction study has shown that average size of metal silver nanoparticles (average area of coherent scattering), determined by Debye–Scherrer method equals to 4–10 nm.

According to transmission electron microscopy, the silver nanoparticles are electron-contrast moderately polydisperse spheres incorporated into a PVT matrix. Their general size ranges 2–26 nm (Figure 5). The narrowest polydispersity of the nanoparticles is observed for nanocomposite 2, prepared using sodium borohydride as a reducing agent, 92% of nanoparticles have the size of 2–6 nm, the dimensions of other nanoparticles (8%) range 6–10 nm (Figure 5a,b). When formaldehyde having a lower reactivity was used as a reducing agent (nanocomposite **6**), the polydispersity of AgNPs becomes wider, the sizes of silver nanoparticles are 2–14 nm (Figure 5c,d).

The incorporation of AgNPs into the PVT matrix decreases the thermostability threshold of silver-containing nanocomposites than the starting PVT: these composite nanomaterials undergo thermal destruction at a temperature of 260 °C and higher. When diluted in water, the solid samples of nanocomposite are decomposed to give solvated complex composite particles with average hydrodynamic diameter of 195 nm (polydispersity is 0.28), which likely represent the globules of separate PVT molecules with an ensemble of silver nanoparticles incorporated therein (Figure 1b).

When analyzing the synthesized nanocomposites **1**–**6**, it was found that nanocomposite **2** is characterized by the formation of smaller and narrowly dispersed silver nanoparticles, which are more evenly distributed in the polymer matrix. Therefore, studies of the antimicrobial action of nanocomposites were carried out precisely using this nanocomposite **2**.

### 3.3. Antimicrobial Activity of Polymeric AgNPs Nanocomposites

Using the technique of microbiologic examination, it was proved that the obtained AgNPs nanocomposites **2** possess a pronounced antimicrobial effect towards different strains of Gram-positive and Gram-negative bacteria (Table 2), while the polymer matrix has no such effect.

The monitoring of the nanocomposite antimicrobial action by transmission electron microscopy technique has shown that in 1 h after the incubation of the nanocomposite with *E. coli*, the starting nanocomposite, which does not contact with the bacteria, is mainly observed. In addition, free silver nanoparticles (initially not associated with the polymer) are discernible (Figure 6A–C). The major number of bacteria remains intact allowing to monitor many dividing forms (Figure 6A,B). However, by the end of the incubation period, some bacteria are associated either with the nanocomposite (silver nanoparticles in the polymer matrix) or free silver nanoparticles (Figure 6C). There with it is shown that a part of bacteria contacting with silver nanocomposite remains unchanged, whereas the other part has different disorders, so bacteria with first signs of cytoplasm homogenization, ones with string-like release of cytoplasm, and selected bacteria with thinned cell walls are indicated.

After incubation of *E. coli* in 10^5^ CFU/mL concentration with the nanocomposite for 1 h, inoculation was performed on a solid nutrient medium. No bacterial growth was registered. However, when testing the interaction of *E. coli* in 10^7^ CFU/mL concentration with a nanocomposite for 1 h, it did not provide a bactericidal effect, the bacteria concentration reduces to 10^4^ CFU/mL.

Within 2 h of the incubation, as before many fragments of the nanocomposite (incorporated in the polymer matrix of silver nanoparticle) which are not associated with microbial cells are visualized. In addition, the initial silver nanoparticles (locating beyond the polymer and microbial cells) are observed. A number of bacteria are preserved, but a small part of ones shows the features of destruction (Figure 6D). Some dead bacteria with attached silver nanoparticles on their membranes are also present. It should be noted that silver nanoparticles associated with microbial cells have smaller sizes than the most representative fraction of nanoparticles of the initial nanocomposite.

Within 24 h after the incubation, the starting nanocomposite (the polymer with incorporated silver nanoparticles) remains still unchanged. A part of this nanocomposite is free, while other part has a contact with bacteria (Figure 6E). In some cases, free polymer (without nanoparticles) associated with the bacteria surface is visualized. At the same time, some still living bacteria (in different degree of preservation) and limited number of dividing bacteria are observed. However, many affected bacteria with membrane destruction of various intensity are identified. For instance, bacteria with content “leaking” to external medium is detected. Moreover, numerous “shadow-like” dead cells with silver nanoparticles attached on their surface are present. In the last cases, the silver nanoparticles are also smaller than the nanoparticles of starting nanocomposite. Noteworthy, in 24 h of the incubation at the 10^5^ CFU/mL initial concentration of *E. coli*, the preserved bacteria are mainly small in size (with string-like leakage of cytoplasm), while at the 10^7^ CFU/mL initial concentration of *E. coli*, the bacteria become hypertrophied, i.e., increased in size.

The bacteriological study has shown the absence of the visible growth of microorganisms on solid media in the samples with *E. coli* concentration of 10^5^ CFU/mL and 10^7^ CFU/mL after 2 and 24 h of incubation with nanocomposite and subsequent plating on solid medium.

### 3.4. Cytotoxicity of Polymeric AgNPs Nanocomposites

The cytotoxicity study of PVT, as well as AgNPs nanocomposite based on PVT in relation to mammalian cells was carried out on a culture of rat peritoneal fibroblasts of the third passage. Throughout the incubation time, with both PVT and PVT-AgNPs at concentrations from 0.125 µg/mL to 16 µg/mL, it was demonstrated that the cells mainly remained viable, did not morphologically change and did not lose the ability to divide and collagen synthesis (Figure 7). Nevertheless, in all the studied samples, individual rounded and detached fibroblasts were visualized, as well as fibroblasts at the initial stages of collagen destruction, while the number of detached cells increased with raising concentration of the added substance.

The use of fluorescent dyes allowed to estimate the percentage of dead cells, and to find out that the detached fibroblasts represent nonviable and propidium iodide stained cells. The number of PI stained cells also depended on the concentration of the active substance added (Figure 7).

Statistical analysis showed a slight significant increase in the number of dead cells by 1.84, 1.77 and 1.94 times compared with the control when exposed to fibroblasts for 24 h with PVT at concentrations of 0.5, 4 and 8 µg/mL, respectively (Figure 8A).

At the same time, incubation of a cell culture with a PVT and silver nanocomposite at concentrations of 4, 8 and 16 µg/mL for 24 h led to an increase in cell death by 1.97, 1.99 and 2.74 times, respectively, compared to control (Figure 8B).

Pairwise comparison of the obtained data on the fibroblast death rate under the influence of PVT and PVT-AgNPs in different concentrations using the Mann–Whitney test showed significant differences only when the cells were incubated with substances at a concentration of 16 μg/mL, while when using other concentrations, the level of cell death did not significantly distinguish for test substances (Figure 8C).

Thus, the impact on the culture of peritoneal fibroblasts with concentrations from 0.125 to 0.25 µg/mL for PVT and from 0.125 to 4 µg/mL for AgNPs nanocomposite can be considered nontoxic, since the level of cell death at these concentrations does not significantly differ from that in the control groups, in addition, the tested substances in the indicated concentrations do not affect the morphology of cells, their ability to divide and synthesize collagen.

## 4. Discussion

Apparently, antimicrobic action of silver-containing nanocomposite can be caused by both of its components (polymer and nanosilver).

In particularly, heterocyclic polymer contained in the nanocomposite is proposed to exhibit certain antimicrobic properties in an aqueous solution due to the protonation of basic triazole heterocycles by water and transformation to polycationic macromolecule. The latter one (due to its positive charges) is potentially capable of cooperatively coupling in an aqueous solution with negatively charged membrane of a microbial cell. Such powerful multipoint electrostatic interaction is actively used in modern polycationic antiseptics [22].

Actually, the association of both free polymer (without nanoparticles) and nanocomposite on its basis with a microbial cell is indicative of the binding with microbe (Figure 6C,E). However, this equilibrium process of the polymer protonation by water does not give highly enough contribution of the charged forms and association with the cells is not therefore general owing to its potential reversibility. This statement is supported by the presence of polymer and nanocomposite on its basis, which are not associated with the cell, in the intercellular space during all monitoring experiment. Test experiments with the polymer, which does not show in the pure state (without silver) noticeable bacteriostatic and bactericidal activity, also prove these observations (Table 2).

In turn, it is a common knowledge that silver (in the form of ions or nanoparticles) is capable of effectively exerting the antimicrobial properties when directly contacting to a microbial cell [23,24,25,26]. In our certain case, the silver nanoparticles in a polymeric matrix either should be transported by this polymer to a microbe with subsequent release on a cell surface (for example, in the result of biodegradation–bio-utilization of polymer by the microbe) or the silver ions, which are in equilibrium with a metal skeleton of the nanoparticles, should migrate from the nanocomposite and also reach the microbial cell [27].

The observation shows that these processes are apparently nonequivalent and extended in time. For instance, despite the above examples of association of silver-containing nanocomposite with the microbial cell, the complete irreversible transport of the nanosilver to a microbe (by polymer) is not carried out. This follows from the fact that the nanocomposite particles, non-associated with the cell, are present in intercellular space during all the monitoring time. In the course of the experiment, the marked biodegradation of vinyl polymer, potentially stable for microbial biodecomposition is also not observed.

The second key stage of antimicrobic action (migration of ionic silver from the nanocomposite to microbial cell) effected also due to a structure of polymer having unshared electron pairs at nitrogen atom of the triazole cycles. The latter ones can form coordination bonds with metal ions, in particular, with silver, thus constantly intercepting and extending in time the migration of silver from polymer. Nevertheless, this migration of silver to the microbial cell does occur and is completed with new nanoclusterization of silver involving structures of the microbial cell. This follows from the observations of still living microbial cells becoming higher electronically contrast owing to accumulation of silver as well as dead cells associated with silver nanoparticles of new nanomorphology, for instance with those having smaller sizes (approximately two times) in comparison with the initial nanocomposite (Figure 6D). The additional evidence of such mutual alteration of the initial nanocomposite and microbial cells is the decrease of silver nanoparticles average sizes (Figure 6D).

Obviously, the resultant suppression (accompanied by the dysfunction of cellular development and formation of the small or hypertrophied forms of the bacteria) as well as destruction of the microbial cell can be caused by the whole complex of the physicochemical and microbiologic phenomena arisen in the course of interaction between the nanocomposite and the microbial cell. This complex includes disorders of physiological functions of the membrane and cytoplasmatic components due to their immediate interaction with silver (especially effective in the case of repeatedly strengthened cooperative effect of multiple ensemble of silver nanoparticles surface atoms, subsequent destruction of the microbial cells membrane under the action of photoinduced free radicals resulting from exposure of environment molecules on the nanoparticles surface) [28,29,30] and additional oxidizing action of bivalent silver cations Ag^2+^ (photoinduced directly from silver atoms of the nanoparticles).

Moreover, an important factor of antimicrobial effect can be remote action of plasmon field of the nanoparticles to membrane potential of the cell through a space giving rise to breach of its physiological properties up to the total depolarization or electric destruction of the membrane (similar to plasmon effects surfaces enhanced infrared absorption (SEIRA), surfaces enhanced Raman scattering (SERS) which also are effectively induced by metal nanoparticles).

At the same time, it is known that silver nanoparticles can be toxic to mammalian cells. There are data evidenced silver nanoparticles can penetrate cells. In particular, Franchi LP et al. (2015) showed that AgNPs were able to cross the plasma membrane and were predominantly found in endocytic vesicles [4]. When incubating human gingival fibroblast cells (CRL-2014) with AgNPs, it was found that AgNPs penetrated the cell membrane and localized inside the mitochondria [31]. In additional it was determined that AgNPs induced loss of cell viability and its ability to proliferation in a dose-dependent manner [18,32,33,34,35].

The effect of the AgNPs size on cytotoxicity was also described. It was shown that 10 nm AgNPs were characterized more efficient cell–particle contact which supports higher intracellular bioavailability of silver than in case of bigger Nps [36,37].

The molecular mechanism of AgNPs-induced cytotoxicity includes activation of oxidative stress [32,38], impairment of the mitochondrial function [18], DNA fragmentation induction [4].

The stabilization of silver nanoparticles using nontoxic polymers promote, in our opinion, reduce the toxicity of silver nanocomposites in relation to mammalian cells while maintaining antibacterial activity. Which extends the application of silver nanoparticles in medicine [39].

## 5. Conclusions

In the study, a new water-soluble polymer nanocomposites with silver nanoparticles (2–14-nm in size) incorporated into a matrix of a water-soluble biocompatible polymer—PVT were synthesized. Using modern methods of analysis, the structural, optical, thermal and morphologic properties of nanocomposites were studied. By the technique of microbiologic research, it was proved that the obtained nanocomposites have a pronounced antimicrobial effect in relation to various strains of Gram-positive and Gram-negative bacteria (notably *E. coli* (ATCC 25922), *P. aeruginosa* (ATCC 27853), *K. pneumonia* (ATCC 700603) and *S. aureus* (ATCC 25923)). In result of transmission electron microscopy research, the antimicrobial effect of the nanocomposite during its incubation with E. coli was found. The initial polymer silver-containing nanocomposite remains unchanged within 24 h after incubation. Part of this nanocomposite is visualized as free, while the other part is found on the surface of bacteria. In some cases, a free polymer (without silver nanoparticles) is also bound to the surface of bacteria. A bacteriological study showed the absence of visible growth of microorganisms on a solid medium in samples with a concentration of *E. coli* of 10^5^ CFU/mL and 10^7^ CFU/mL after 2 and 24 h of incubation.

At the same time, the nanocomposite showed low toxicity with respect to peritoneal fibroblasts at a concentration of 0.125 to 4 µg/mL. Thereby low toxicity combined with antibacterial activity makes the developed material promising for use in biomedical purposes.

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
