# Peer review of "NonToxic Silver/Poly-1-Vinyl-1,2,4-Triazole Nanocomposite Materials with Antibacterial Activity"

_nanomaterials, 2020, doi:10.3390/nano10081477_

Round 1

Reviewer 1 Report

The authors present an interesting article entitled "NON-TOXIC SILVER / POLY-1-VINYL-1,2,4-2 TRIAZOLE NANOCOMPOSITE MATERIALS 3 WITH ANTIBACTERIAL ACTIVITY"

The abstract is clear and concise.
The introduction is clear and concise.
The experimental section is clear.
The results and discussion section is ok.
The conclusion is clear and concise.
The references are balanced.

Edits:

Title: Please change ALL CAPS "NON-TOXIC SILVER / POLY-1-VINYL-1,2,4-2 TRIAZOLE NANOCOMPOSITE MATERIALS 3 WITH ANTIBACTERIAL ACTIVITY" to "Title Case".

Abstract:
Please change "Gram-positive Gram-negative bacteria" to read "Gram-positive and Gram-negative bacteria,"

Line 51: "(summarized" there should be a closed bracket somewhere ")"

Table 1, sample 4, Solubility in Н2О = +–? I guess it should be either + or -.

2.8. "in concordance with the protocol approved by the Institutional Animal Care and Use Committee of the Irkutsk Scientific Center of Surgery and Traumatology." do you have an approval number to cite?

Line 324: "Within 2 of the incubation" should probably read "Within 2h of incubation"

Line 431: "(especially" there should be a closed bracket somewhere ")"

Figure 7: images need scale bars, and the legend needs to note the length of the scale bars.Figure 8: please align all images vertically. A on top of B, B on top of C.

Author Response

We are incredibly grateful to you for the great work with our manuscript and very useful comments and suggestions.

The following are the changes we made to the original manuscript and our responses to comments:

Title: Please change ALL CAPS "NON-TOXIC SILVER / POLY-1-VINYL-1,2,4-2 TRIAZOLE NANOCOMPOSITE MATERIALS 3 WITH ANTIBACTERIAL ACTIVITY" to "Title Case".

Answer:

Corrections made on Lines 2-3.

Abstract:
Please change "Gram-positive Gram-negative bacteria" to read "Gram-positive and Gram-negative bacteria,"

Answer:

Corrections made on Line 14.

Line 51: "(summarized" there should be a closed bracket somewhere ")"

Answer:

Corrections made on Line 53.

Table 1, sample 4, Solubility in Н2О = +–? I guess it should be either + or -.

Answer:

Corrections made in Table 1, sample 4.

2.8. "in concordance with the protocol approved by the Institutional Animal Care and Use Committee of the Irkutsk Scientific Center of Surgery and Traumatology." do you have an approval number to cite?

Answer:

Corrections made on Line 194.

Line 324: "Within 2 of the incubation" should probably read "Within 2h of incubation"

Answer:

Corrections made on Line 322.

Line 431: "(especially" there should be a closed bracket somewhere ")"

Answer:

Corrections made on Line 434.

Figure 7: images need scale bars, and the legend needs to note the length of the scale bars.

Answer:

Corrections made on Lines 354-355, 360-361.

Figure 8: please align all images vertically. A on top of B, B on top of C.

Answer:

Corrections made on Line 366.

Sincerely,

Prof. Irina Shurygina

Reviewer 2 Report

This is an interesting research article on the identification of novel silver/poly-1-vinyl-1,2,4-triazole nanocomposite materials with antimicrobial activity against E. coli. Novel antimicrobials are urgently needed, therefore this paper is reporting very interesting data. However, there is a number of limitations that should be corrected before its publication:

The English language should be proof-read by a native speaker. In addition, there are parts of the text that are unclear. E.g. in the abstract: Gram positive Gram negative, etc.

Line 58, why is that? Please explain this point with more details.

Lines 150-151, a species name should not be abbreviated the first time that is mentioned in the text. In addition, you should add here some more information about this particular set of strains. E.g. are these reference strains for antibiotic testing?

Line 174, this is unclear, please explain this with more details.

Table 2, it seems that this compound is highly active against E. coli but not against any other Gram-negative or Gram-positive bacteria tested here. Authors should discuss in detail this observation, it is quite unusual to identify a compound based on Ag with such a specificity for a particular bacterial species. This may have implications on its mode of action.

Figure 6, where are the bacteria mentioned in the text? I cannot identify any E. coli cells (which is bacillar) in these microphotographs.

Lines 320 and 343, are authors using here a TEM to evaluate bacterial viability? If so, this should be mentioned here and explained with much more detail. 

Figure 7, authors have measured cell viability by using FDA and PI staining. The microphotographs included here are quite difficult to interpret. Authors should repeat this experiment with Annexin V FITC and PI staining to get a better picture on the viability of the cells treated with these compounds since PI staining is reporting mainly necrotic cells. It would be good to add a positive control of cell death induction.  

Author Response

We are incredibly grateful to you for the great work with our manuscript and very useful comments and suggestions.

The following are the changes we made to the original manuscript and our responses to comments:

The English language should be proof-read by a native speaker. In addition, there are parts of the text that are unclear. E.g. in the abstract: Gram positive Gram negative, etc.

Answer:

Corrections made on Line 15. We ask the editorial staff to carry out professional editing of the text.

Line 58, why is that? Please explain this point with more details.

Answer:

It is literature data. We insert part from reference - Corrections made on Lines 59-60.

Lines 150-151, a species name should not be abbreviated the first time that is mentioned in the text. In addition, you should add here some more information about this particular set of strains. E.g. are these reference strains for antibiotic testing?  

Answer:

Abbreviations in Results are given after the first mention in the text (Materials and Methods, lines 84-86). These are reference strains - corrections made on Line 84.

Answer:

Line 174, this is unclear, please explain this with more details.

Corrections made on Line 172: After incubation of inoculum in 35 °C, the lowest concentration of the test substance with no bacterial growth was determined. This concentration corresponds to as MBC.

Table 2, it seems that this compound is highly active against E. coli but not against any other Gram-negative or Gram-positive bacteria tested here. Authors should discuss in detail this observation, it is quite unusual to identify a compound based on Ag with such a specificity for a particular bacterial species. This may have implications on its mode of action.

Answer:

Thanks for advice. We added the text to «Discussion»:

When studying the toxicity of nanocomposites to bacterial strains, the test strain E. coli ATCC 25922 showed significantly greater sensitivity to the nanocomposite compare average inhibitory concentrations of 4-8 μg / ml for other strains. Since the polymer matrix had not the antibacterial effect for this strain, it can be assumed that this microorganism actively interacts with the nanocomposite. The result of this interaction is the exposure of silver nanoparticles to the microbial wall. The study of this phenomenon requires additional research. (Lines 442-447).

Figure 6, where are the bacteria mentioned in the text? I cannot identify any E. coli cells (which is bacillar) in these microphotographs.

Answer:

Our laboratory has developed a methodology for assessing the interaction of specific bacterial strains with the studied nanocomposites. It is set out in section "2.7. Incubation of nanocomposite with E. coli for electron microscopic study», as well as in the article Shurygina, I.A.; Fadeeva, T.V.; Umanets, V.A.; Shurygin, M.G.; Grigoriev, E.G.; Sukhov, B.G.; Ganenko, T.V.; Kostyro, Y.A.; Trofimov, B.A. Bactericidal action of Ag(0)-antithrombotic sulfated arabinogalactan nanocomposite: coevolution of initial nanocomposite and living microbial cell to a novel nonliving nanocomposite. Nanomedicine 2011, 7, 827-833. doi: 10.1016/j.nano.2011.03.003. To exclude the influence of the components of the medium on the results of the interaction of the nanocomposite and the bacteria, a “poor” nutrient medium — phosphate buffer — is used. A standard bacterial strain also washed from the medium before the study. This preparation determines the atypical morphological of bacteriae.

Lines 320 and 343, are authors using here a TEM to evaluate bacterial viability? If so, this should be mentioned here and explained with much more detail. 

Answer:

Lines 320, 343  - The study was conducted using the standard microbiological method for bacterial viability - growth on Mueller–Hinton agar.

Figure 7, authors have measured cell viability by using FDA and PI staining. The microphotographs included here are quite difficult to interpret. Authors should repeat this experiment with Annexin V FITC and PI staining to get a better picture on the viability of the cells treated with these compounds since PI staining is reporting mainly necrotic cells. It would be good to add a positive control of cell death induction.  

Answer:

FDA and PI stains are widely used by many researchers to identify living and dead cells. For example,

Cellular architecture and migration behavior of fibroblast cells on polyhydroxyoctanoate (PHO): A natural polymer of bacterial origin / T. Witko, D. Solarz, K. Feliksiak, Z. Rajfur, M. Guzik – Biopolymers, 2019; e23324.

Polyhexamethyleneguanidine phosphate induces cytotoxicity through disruption of membrane integrity / J. Song, K.J. Jung, S. Yoon, K. Lee, B. Kim – Toxicology, 2019, Vol. 414, 35–44.

In vitro and in vivo effects of insulinproducing cells generated by xenoantigen free 3D culture with RCP Piece / T. Ikemoto, R. Feng, S. Iwahashi, S. Yamada, Y. Saito, Y. Morine, S. Imura, M. Matsuhisa, M. Shimada – Scientific Reports, 2019, Vol. 9:10759. doi: 10.1038/s41598-019-47257-7

Protection of mouse pancreatic islet function by co culture with hypoxia pre treated mesenchymal stromal cells / C. Xiang, Q. P. Xie – Molecular Medicine Reports, 2018, Vol. 18: 2589-2598.

Yahaya ES, Cordier W, Steenkamp PA, Steenkamp V (2020) Protective effect of Erythrina senegalensis sequential extracts against oxidative stress in SC-1 fibroblasts and THP-1 macrophages. J Pharm Pharmacogn Res 8(4): 247–259.

Assessment of Neuronal Viability Using Fluorescein Diacetate-Propidium Iodide Double Staining in Cerebellar Granule Neuron Culture / L. Jiajia, M. Shinghung, Z. Jiacheng, W. Jialing, X. Dilin, H. Shengquan, Z. Zaijun, W. Qinwen, H. Yifan, C. Wei – Journal of Visualized Experiments May, 2017, 10 (123):55442.

Effects of different concentrations of type-I collagen hydrogel on the growth and differentiation of chondrocytes / D. Hu, X. Shan – Experimental and Therapeutic Medicine, 2017, Vol. 14 (6).

Isolation and characterisation of human gingival margin-derived STRO-1/MACS1 and MACS2 cell populations / K.M. Fawzy El-Sayed, S. Paris, C. Graetz, N. Kassem, M. Mekhemar, H. Ungefroren, F. Fandrich, C. Dorfer – International Journal of Oral Science, 2014, Vol. 7, 80–88.

Polydimethylsiloxane Films Modified with Chitosan/Pectin Multilayers as Scaffolds for Mesenchymal Stem Cells /V. I. Kulikouskayaa, I. V. Pariboka, S. V. Pinchukb, A. N. Kraskouskia, I. B. Vasilevichb, K. A. Matievskib, V. E. Agabekova, I. D. Volotovskib – Applied Biochemistry and Microbiology, 2018, Vol. 54, No. 5, pp. 468–473.

Therefore, we found it possible to apply these methods in our work.

Unfortunately, in the near future due to the epidemiological situation with COVID-19, we are not able to work in the laboratory. All employees, by order of the Institute’s management, perform work remotely.

Sincerely,

Prof. Irina Shurygina

Reviewer 3 Report

This paper well described that the properties of non toxic silver nanocomposite allow to use antimicrobial activity. The authors show novel silver nanocomposite have a pronounced antimicrobial effect on gram positive and negative bacteria. And also, there are shown low toxicity against mammalian cell even though including silver nanoparticles. It is very interesting to reduce the toxicity for mammalian cell membrane.

Then, I would like to accept this paper to be published in Nanomaterials at this time.

I would like to comment on some below

  1. Page2, line 51, Missing the parenthesis somewhere.
  2. In Fig. 1(b), What is the size of AgNPs nanocomposite size? I think the author mistake the number (1) and (2) at Fig.1(b)
  3. Page 9, Line 304, It is necessary to distinguish the gram positive and negative bacteria at Table 2.
  4. 6, Put in indicator such as arrow for AgNPs and E. coli
  5. Page 11, line 338-339 How much reduce the size of silver nanoparticles compare with starting nanocompsite?
  6. In Fig. 8, is there any cytotoxicity data for AgNPs alone?
  7. Page 15, line 440, Change the order of sentence (surface Enhanced Infrared Absorption(SEIRA), and -------(SERS))
  8. Page15, line 477, E.coli

Author Response

We are incredibly grateful to you for the great work with our manuscript and very useful comments and suggestions.

The following are the changes we made to the original manuscript and our responses to comments:

Page2, line 51, Missing the parenthesis somewhere.

Answer:

Corrections made on Line 53.

In Fig. 1(b), What is the size of AgNPs nanocomposite size? I think the author mistake the number (1) and (2) at Fig.1(b)

Answer:

In Fig. 1(b) the size of AgNPs nanocomposite is 195 nm (number 2),  the size of PVT is 292 nm (number 1), A decrease in the hydrodynamic diameters of AgNPs nanocomposite particles to 195 nm compared with the diameter of the initial PVT macromolecules (292 nm) is due to the fact, that during the synthesis silver nanoparticles are involved not only in intramolecular coordination, but also in intermolecular crosslinking. This leads to increase the density ("compression") of macromolecular globules and a decrease in the hydrodynamic diameters of nanocomposite particles.

Page 9, Line 304, It is necessary to distinguish the gram positive and negative bacteria at Table 2.

Answer:

Corrections made in Table 2.

Put in indicator such as arrow for AgNPs and E. coli

Answer:

Corrections made on Line 316

Page 11, line 338-339 How much reduce the size of silver nanoparticles compare with starting nanocomposite?

Answer:

Thanks for the nice question. We will calculate statistics for nanocomposites in the future. Unfortunately, this work will not be completed in a short time.

In Fig. 8, is there any cytotoxicity data for AgNPs alone?

Answer:

Previously, many studies have shown high cytotoxicity for AgNPs alone. This paper presents the results of studies of Ag  nanocomposites with polymer. We did not obtain isolated silver nanoparticles without a polymer matrix and did not study their cytotoxicity due to the wide coverage of this issue in previously published studies.

Page 15, line 440, Change the order of sentence (surface Enhanced Infrared Absorption(SEIRA), and -------(SERS))

Answer:

Corrections made on Lines 440-441.

Page15, line 477, E. coli

Correction made on Line 472.

Sincerely,

Prof. Irina Shurygina

Round 2

Reviewer 2 Report

The revision of the first draft has marginally improved the manuscript. My understanding is that MDPI provides a paid service for the editing of the English language, and that this should be done before submission: https://www.mdpi.com/authors/english

The part added to the discussion on E. coli does not improve the text at all, quite the contrary. Authors should provide an explanation of their results, otherwise these data could be interpreted in many alternative ways. For instance, the data reported here could be due to an artefact, which is a plausible explanation to the fact that E. coli is the only bacterial species relevant for the conclusions of this research article.  

Figure 6 is a major problem. Authors did not include a negative control and what they describe as bacteria could be anything. In fact, these could be artefacts. If a negative control of untreated bacteria is not added, this figure should be removed from the text.

Lines 318 and 342, are authors suggesting here that differences in bacterial viability after 1 or 2 hours of incubation on agar plates can be observed by naked eye?

Figure 8, there are random red spots on the picture that may or may not correspond to dead cells. In addition, the number of dead cells is minimal in panel C. These pictures may not be representative of what authors are reporting on Figure 9B, in which authors are reporting a 10% of cell death at 16 ug/ml. Authors should either improve Figure 8 or remove it.

Author Response

Dear Ms. Katherine Bian,

We are very grateful to you and reviewers for the great work with our manuscript and very useful comments and suggestions.

We hereby present the manuscript entitled “NON-TOXIC SILVER / POLY-1-VINYL-1,2,4-TRIAZOLE NANOCOMPOSITE MATERIALS WITH ANTIBACTERIAL ACTIVITY” revised in accordance with all the comments and advice of reviewers.

The following are the changes we made to the original manuscript and our responses to comments made by the referees.

The revision of the first draft has marginally improved the manuscript. My understanding is that MDPI provides a paid service for the editing of the English language, and that this should be done before submission: https://www.mdpi.com/authors/english

The part added to the discussion on E. coli does not improve the text at all, quite the contrary. Authors should provide an explanation of their results, otherwise these data could be interpreted in many alternative ways. For instance, the data reported here could be due to an artefact, which is a plausible explanation to the fact that E. coli is the only bacterial species relevant for the conclusions of this research article.  

Answer:

We have not studied the mechanisms of the toxic effect of the nanocomposite on different types of bacteria. Previous text insertion removed.

Figure 6 is a major problem. Authors did not include a negative control and what they describe as bacteria could be anything. In fact, these could be artefacts. If a negative control of untreated bacteria is not added, this figure should be removed from the text.

Answer:

  1. coli does not always exist in the classic rod-shaped form. Depending on the vital conditions, they change their shape, which was thoroughly studied in the 60-70s of the last century. It is known that transition of bacteria to cell wall deficient L-forms is a mechanism for survival under unfavorable conditions. Transmission electron microscopy demonstrated conversion from classical rod to polymorphic L-form shape morphology of E. coli. As a very good example, we can cite the article [Nadya Markova et al. Survival of Escherichia coli under lethal heat stress by L-form conversion. Int J Biol Sci. 2010 Jun 9; 6 (4): 303-15. doi: 10.7150 / ijbs.6.303. https://pubmed.ncbi.nlm.nih.gov/20582223/]. The authors demonstrated the transformation of bacteria into the L-form, and the reason for the transformation is the incubation of E. coli with minimal salt broth (K2HPO4, KH2PO4, MgSO4, (NH4) 2SO4, pH 7.2). The differences in the image of L-forms with our example are that in the above article, Markova N. et al. (2010) used ultrathin sections of transforming E. coli cells. We have visualized whole bacteria in our work.

The morphology of whole bacteria in L-form by TEM can be seen in the article https://jb.asm.org/content/jb/90/5/1467.full.pdf (unfortunately of low quality).

We included the control in Figure 6F (Line 314) to illustrate bacteria grown in a poor environment and not exposed to the nanocomposite.

Lines 318 and 342, are authors suggesting here that differences in bacterial viability after 1 or 2 hours of incubation on agar plates can be observed by naked eye?

Answer:

The word "incubation" in our text means the interaction of bacteria with the nanocomposite. We incubated bacteria with nanocomposite for 1, 2 or 24 h. Next, we used classical microbiological studies for the viability of bacteria with inoculation on a solid medium. As suggested by the standard procedure, the culture was grown on solid media for 1 day. After that, the conclusion about the viability was made by the presence or absence of colonies on Mueller–Hinton agar, of course, with the naked eye.

We've changed the text a bit for clarity:

After incubation of E. coli in 105 CFU/mL concentration with the nanocomposite for 1 h, inoculation was performed on a solid nutrient medium. No bacterial growth was registered. However, when testing the interaction of E. coli in 107 CFU/mL concentration with a nanocomposite for 1 h, it did not provide a bactericidal effect, the bacteria concentration reduces to 104 CFU/mL (Lines 318-321).

The bacteriological study has shown the absence of the visible growth of microorganisms on solid media in the samples with E. coli concentration of 105 CFU/mL and 107 CFU/mL after 2 and 24 h of incubation with nanocomposite and subsequent plating on solid medium (Line 343).

Figure 8, there are random red spots on the picture that may or may not correspond to dead cells. In addition, the number of dead cells is minimal in panel C. These pictures may not be representative of what authors are reporting on Figure 9B, in which authors are reporting a 10% of cell death at 16 ug/ml. Authors should either improve Figure 8 or remove it.

Answer:

In Fig. 7, we presented the 24 h fibroblasts incubation for control, PVT 16 μg / ml, nanocomposite AgNPs (PVT-Ag) 16 μg / ml. To minimize the subjective factor, all images were acquired and processed automatically (many tens for each sample for full coverage of the plate). This method, like others, is not ideal, but according to the principles of detection, it is the closest to flow cytometry with the difference in the possibility of dynamic observation of the cell culture. Phase contrast, fluorescent staining with FDA (green) and PI (red) are presented in rows from the same FoV. I hope this presentation will be more descriptive (Line 356).

Sincerely,

Irina Shurygina
